# Towards Predicting Gut Microbial Metabolism: Integration of Flux Balance Analysis and Untargeted Metabolomics

**DOI:** 10.3390/metabo10040156

**Published:** 2020-04-17

**Authors:** Ellen Kuang, Matthew Marney, Daniel Cuevas, Robert A. Edwards, Erica M. Forsberg

**Affiliations:** 1Department of Chemistry and Biochemistry, San Diego State University, San Diego, CA 92182, USA; 2Department of Biomedical Informatics, San Diego State University, San Diego, CA 92182, USA; 3Viral Information Institute, San Diego State University, San Diego, CA 92182, USA; 4Department of Biology, San Diego State University, San Diego, CA 92182, USA

**Keywords:** metabolomics, flux balance analysis, multiomics, bioinformatics, mass spectrometry, microbiome

## Abstract

Genomics-based metabolic models of microorganisms currently have no easy way of corroborating predicted biomass with the actual metabolites being produced. This study uses untargeted mass spectrometry-based metabolomics data to generate a list of accurate metabolite masses produced from the human commensal bacteria *Citrobacter sedlakii* grown in the presence of a simple glucose carbon source. A genomics-based flux balance metabolic model of this bacterium was previously generated using the bioinformatics tool PyFBA and phenotypic growth curve data. The high-resolution mass spectrometry data obtained through timed metabolic extractions were integrated with the predicted metabolic model through a program called MS_FBA. This program correlated untargeted metabolomics features from *C. sedlakii* with 218 of the 699 metabolites in the model using an exact mass match, with 51 metabolites further confirmed using predicted isotope ratios. Over 1400 metabolites were matched with additional metabolites in the ModelSEED database, indicating the need to incorporate more specific gene annotations into the predictive model through metabolomics-guided gap filling.

## 1. Introduction

Studies of the gut microbiome have been shown to have a major impact on health and the risk for disease. Predicting metabolic output in the gut microbiome would be particularly beneficial in understanding how different nutrient sources influence metabolite production. Microbial metabolic products have been shown to be involved in intercellular communication and immune response mechanisms in the gastrointestinal tract [1,2]. It would, therefore, be beneficial to predict which bacterial species are capable of producing bioactive metabolites under different nutrient sources. Although there is no method that can directly model the gut microbiome as an entire system [3,4], flux balance analysis [5] does allow for the development of metabolic models for individual species. Developing a predictive model in microorganisms requires a sequenced genome and knowledge of gene function to annotate the genome, and thus which proteins are translated. Gene annotations are based on sequence similarity to known and conserved gene sequences. In cases where sequence similarity is low to none, a process known as gap-filling is performed to make an educated guess on gene function. By monitoring experimental growth under a variety of carbon, nitrogen, and sulfur sources, biomass production can be an indicator of whether a particular set of metabolic pathways or transport functions exists in the genome, regardless of known sequence similarities [6]. Metabolism of these nutrient sources produces diverse metabolites that cannot be confirmed with phenotypic growth curve assays, nor a predictive method such as flux balance analysis. In addition, there is currently no way of easily corroborating predicted biomass with the actual metabolites being produced under specific nutrient conditions. Our goal is to compare a genome-based metabolic model with untargeted metabolomic data to confirm the metabolome produced by a bacterium. Although applying untargeted metabolomic data to systems biology has been a challenge in the field [7], the advent of sensitive, high-resolution mass spectrometers [8] and recent metabolomic bioinformatics tools [9,10,11,12,13,14] provides a great opportunity to improve multi-omic integration techniques to understand microbial metabolism.

Mass spectrometry-based metabolomic data are already being used to assist in characterizing metabolic models by comparing the metabolites produced in cell cultures to expected biomass output via flux balance analysis (FBA) [15,16,17,18,19]. Although there are some excellent bioinformatic tools available for integrating metabolomic data into flux-balance-generated metabolic models [15], they require a significant amount of preliminary experimental work using multiple targeted quantitative metabolomics assays, often requiring use of expensive isotopically labeled reagents. The MS_FBA bioinformatics script illustrated in Figure 1 takes advantage of data generated from a simple untargeted metabolomics workflow. Instead of performing targeted analysis on a list of hundreds to thousands of predicted metabolites, we take advantage of information-dense untargeted metabolomic data to identify metabolites that are biosynthesized by bacterial cells. The script was not designed to obtain absolute quantitative data, but rather to confirm production or consumption of metabolites, thereby giving a broad picture of the metabolome; in this case, we utilize the internal metabolites to simplify the analysis, but any source of bacterial growth over the course of its curve can be utilized. Using a time course of bacterial growth, untargeted metabolomic data are produced over the growth curve to differentiate between metabolites synthesized within the cell or depleted through salvage and transport pathways. MS_FBA then compares the untargeted feature list to a predicted metabolite list based on the FBA of the microbe of interest. These can be either a predicted FBA compound list or an entire database, such as the ModelSEED database [20]. Using predicted compounds from the metabolic model is preferable, since it will take significantly less time and reduce the false positive rate by reducing the number of spurious observations. The ModelSEED database becomes important when assessing the accuracy of the metabolic model and potentially unpredicted metabolites. MS_FBA uses a PyFBA [21] generated compound list to compare with metabolite mass-to-charge ratios (*m*/*z*) of significantly changing features detected from XCMS Online [22] results. Metabolite identities are also assessed using isotope ratios found within the raw data. This application creates a simple and efficient way to evaluate metabolic models, thereby validating predicted gene annotations based on gap-filling methods. *Citrobacter sedlakii* (*C. sedlakii*), having well-established metabolic models [6,23], was used to illustrate how the R script, MS_FBA, can be used to verify a metabolic model using untargeted metabolomic data and assess metabolites that have been left out of the predicted model based on incomplete gap-filling.

## 2. Results

PyFBA [21] was used to generate a metabolic model for *C. sedlakii* in minimal media supplemented with 0.2% glucose. The resulting metabolite list was imported into MS_FBA and stored as a dataframe for integration with high-resolution liquid chromatography mass spectrometry (LCMS) data. MS_FBA was implemented on both the reverse phase (RP) and hydrophilic interaction liquid chromatography (HILIC) datasets. After comparison with the metabolic model, a graphical output was generated to illustrate matches between XCMS Online [24] significant features and the predicted metabolites list (Figure 2). These features either increase or decrease over the course of the growth curve measured at mid-exponential, late-exponential, and stationary phases. MS_FBA also generates a graph that shows all the significant features alone, as well as the features that were not matched to the metabolic model. These are illustrated in Appendix A for the reverse phase run performed in positive mode.

The metabolite features identified as significantly changing between time points on the growth curve were matched to a PyFBA predicted metabolite list based on a *C. sedlakii* metabolic model, using glucose as a carbon source. Isotope matches and adducts were readily validated in the raw data and in the XCMS feature results, as demonstrated with glutamic acid from the RP data in Figure 3. MS_FBA uses a preset list of adducts (Appendix A) that it searches for within a given peak group to give higher confidence in metabolite identity. Glutamic acid was accurately matched within that peak group to a sodium adduct, a neutral loss of a water molecule, and the dimer. In addition, there is a loss of a carboxylic acid group and a trimer that MS_FBA did not predict to be related to glutamate, although upon inspection of the raw data are clearly related peaks. Numerous anticipated metabolites, such as glutamate, were shown to have a significant increase from mid- to late-exponential growth phases, while decreasing from late to stationary phases, indicating that the metabolites were being consumed by salvage pathways after nutrients in the media were consumed. This is depicted in the Figure 3 inset with an XCMS generated box and whisker plot of mid, late, and stationary phase sample classes.

In addition to the predicted metabolite list, the ModelSEED database was also input as a full list of metabolites to assess compounds in the metabolomic data that were not matched to the PyFBA model. The RP data were run in positive mode and the HILIC data in negative mode to achieve better separation of both polar and nonpolar metabolites and to improve detection of analytes that are not readily ionizable in both MS polarities. The results of the MS_FBA analyses are reported in Table 1 and Figure 4. A total of 253 compounds were detected in both LCMS runs (Appendix A) out of the 699 expected compounds. Of the matched compounds, 109 compounds were found exclusively in the HILIC data, 110 compounds exclusively in RP, and 35 compounds in both.

### 2.1. Reverse Phase Results

The XCMS Online results identified 1210 features from the reverse phase positive mode run that were significantly changed using a *P*-value threshold of 0.05. Of those features, 135 were matched to expected compounds list from PyFBA (Appendix A); 23 of those were also isotope ratio matches (Figure 4). When the 1210 features were compared to the ModelSEED database, 846 features were matched with accurate mass and 107 of those were isotope ratio matches. Each feature has the possibility of matching with multiple compounds in the database if the compounds have the same mass within a 5 ppm deviation. The 135 features from the reverse phase run matched to 210 potential compounds from the PyFBA list, while the 846 features matched to 3242 potential compounds from the ModelSEED database.

### 2.2. HILIC Results

In comparison to the reverse phase data, the XCMS Online results identified 1560 features from the HILIC negative mode data that were under the *P*-value threshold of 0.05. Of those 1560 features, 118 were matched to the expected compounds list from PyFBA (Appendix A), 28 were isotope ratio and accurate mass matches, while 90 features matched accurate mass alone. When the 1560 features were compared to the ModelSEED database, 621 features were matched— 90 isotope ratio and accurate mass matches and 531 accurate mass only matches. Each feature has a chance of matching to multiple compounds in each database or list used if the compounds are isomers. The 118 features from the HILIC run matched to 148 compounds from the PyFBA list, while the 621 features matched to 1983 compounds from the ModelSEED database.

MS_FBA also generates a list of the matched compounds (Appendix A) describing the compound name, the feature ID that was given in the XCMS Online results, the observed *m*/*z* value of the feature, its retention time, the peak group in which it elutes, the isotope ratio difference, whether it met the isotope ratio criteria to the identified compound, and the identified adduct. Adducts are considered within 5 ppm of the monoisotopic mass of the predicted metabolite plus the mass of the ionizing species, as provided in Appendix A, for either positive or negative mode. Peak groups are features that have eluted within the same time window and indicate potential adducts or co-eluting species. In addition to this, there are some feature IDs that are repeated in these tables because there are either structural isomers or enantiomers. The best way to alleviate this issue is to confirm the identity of the compound using a standard to confirm both retention time and tandem MS fragmentation pattern. In this particular study of *C. sedlakii* grown in glucose media, there were 210 potential metabolite matches to 135 features in reverse phase positive mode and 148 potential matches to 118 features in HILIC negative mode when compared to the metabolic model.

## 3. Discussion

A summary of the matched feature results for both the RP and HILIC datasets is presented in Figure 4. The metabolic model generated 699 metabolites that would be feasibly detected in an LCMS system (i.e., metal ions and low molecular weight molecules were removed, such as CO_2_ and NH_3_). The identification of feature adducts assists in positively identifying features, particularly in the cases where there is an isotope match. In the case of glutamic acid (Figure 3), not all neutral losses were identified in the peak group. Along with the [M+H]^+^ peak, MS_FBA correctly identified the water neutral loss, the sodium adduct, and the dimer. It did not detect the loss of the carboxylate group or the trimer. The adduct list (Appendix A) could easily be adjusted to include the trimer; however, identifying all neutral losses with a single adduct list is not feasible, since metabolites have different functional groups susceptible to in-source fragmentation. Future versions of MS_FBA must include dynamic algorithms to detect neutral losses based on the predicted structure of the parent mass. In the RP dataset, e a total of 135 features were matched to the predicted metabolite list, 23 of which were isotope ratio matches. In the HILIC dataset, a total of 118 features were matched, with 28 of those features having isotope ratio matches. The low percentage of isotope ratio matches is likely a result of low abundance peaks, where the M+1 peak was not detected and was below the signal-to-noise ratio of the instrument. In comparison to the total features from the XCMS Online results (Table 1), it is clear there are more metabolites being produced than are detected and matched to the metabolic model described, particularly with the significant increase in isotope ratio matches. Although MS_FBA does not describe the false discovery rate, it is expected to be rather high, with only five biological replicates. However, the statistical analysis between the three time points is performed coupled with the biological feature check in order to compensate for this.

In total, 35 out of 253 features were identified as being the same in both RP and HILIC datasets. Reverse phase chromatography typically provides coverage for non-polar metabolites, whereas HILIC provides polar metabolite selectivity [25]. These results demonstrate the importance of using multiple chromatographic modes and MS polarities to enhance metabolite coverage. With a total number of 218 unique features being matched to the predicted PyFBA metabolic model of 699 metabolites, we still obtained less than 30% coverage in our model, keeping in mind that some of these features are adducts and correlate to single metabolites. This is where the ModelSEED database can be utilized to see more realistic potential matches. Indeed, it would be better to see more isotope ratio matches to improve confidence in this initial identification. Although the cell density for *C. sedlakii* in 0.2% glucose was exceptionally low (0.2 to 0.5 OD600), we took this as a challenge to see how low our metabolome detection could get, as well as to generate an original yet simple set of data to use with the MS_FBA script. This had an obvious impact, with only 41% of compounds matched to the PyFBA expected list in both the combined RP and HILIC datasets. Considering the cell density was this low, the amount of coverage the MS_FBA script was able to obtain was surprising. However, optimizing the growth portion of the experimental procedure will help when applying MS/MS validation methods in our algorithms.

Although the overall signal intensity and metabolite production were lacking due to the low glucose concentration, there were hundreds more features that had potential matches with the ModelSEED database compared to the PyFBA generated metabolic model. This indicates that the gap filling methods are not sufficient to predict all the metabolic pathways and transport mechanisms that are actually present in the bacterium. Gap filling [26] in PyFBA provides additional enzymatic and transport reactions based on known essential reactions, user-supplied phylogentic information, and analysis of incomplete subsystems to identify reactions and missing enzymes based on metabolites predicted in the model. To prevent an excess of additional reactions that would produce an inaccurate model, a gap generation step is added. This part of the algorithm systematically bisects the added reactions and tests for growth. Reactions are removed until the predicted biomass matches the phenotypic growth curve. Even with our 0.2% glucose data, it is apparent that too many reactions were removed, as more metabolites are observed using the ModelSEED database than those predicted from the PyFBA model. These data will be used in the next version of MS_FBA to reiterate the metabolic model to include reactions that produce observed metabolites. In addition, an extra MS/MS validation step will also be employed to more accurately confirm metabolite identities by including data-dependent fragmentation in the untargeted workflow. This preliminary work with a simple glucose nutrient source is important in developing the workflow for specificity to untargeted data, which are easy to produce but have a high density of information. The next phase will involve modeling more complex mixtures, including Lysogeny Broth (LB) media. If we wish to move towards modeling metabolism in the gastrointestinal tract, we first need to accurately predict whether metabolites are produced or consumed. This first version of MS_FBA has developed an essential tool for identifying metabolites in an untargeted manner that will improve our gap-filling algorithms to produce more accurate metabolic models.

## 4. Materials and Methods

### 4.1. Bacterial Culturing and Sample Preparation

To compare raw metabolomics data to metabolic models generated via flux balance analysis, we grew *C. sedlakii* as a model system, on which the metabolomics workflow was performed. Five biological replicates of *C. sedlakii* cultures were prepared by upscaling the same workflow used to generate the genomic model [23]. From a glycerol stock obtained from the Biology Department stockroom at San Diego State University, a 10 mL overnight growth in tryptone yeast glucose (TYG) media was prepared aerobically by inoculating 1 μL of the glycerol stock. The following day, the overnight growth was streaked onto TYG agar plates to obtain individual colonies. Five colonies were picked and individually grown in 10 mL of TYG media overnight. This overnight growth was then washed three times with sterile phosphate buffered saline (PBS) for transfer to minimal media. The washed pellet was resuspended in 1 mL of PBS, then 2 μL of each pellet was inoculated into 10 mL of modified 3-morpholinopropan-1-sulfonate (MOPS) broth with 0.2% glucose. The modified MOPS media consisted of the following: 1X MOPS (40 mM MOPS + 10 mM Tricine), 0.4% glycerol, 9,5 mM NH_4_Cl, 0.25 mM NaSO_4_, 1.0 mM MgSO_4_, 1.32 mM K_2_HPO_4_, 10mM KCl, 0.5 μM CaCl_2_, 5 mM NaCl, and 6 μM FeCl_3_ [23]. All cultures were incubated at 37 °C and cultures in liquid media were incubated on an orbital shaker at 250 rpm. Cell densities were ranged from 0.2 to 0.5 from mid-log phase to stationary phase. The choice of 0.2% glucose was taken as a direct scale-up from the original plate-based assays used to generate the metabolic model.

Each of the five biological replicates had three replicate samples for three different extraction time points. The negative growth control was inoculated with 2 μL of the sterile wash buffer to ensure sterility during the wash step and was also prepared in triplicate, giving a total of 18 samples. The bacteria were grown to mid, late, and stationary phases before each set of samples (1 negative control and 5 biological replicates) were quenched and extracted. Extraction of the internal metabolites [27] was performed by first pelleting the cells at 4000 rpm for 15 min at 4 °C and removing the supernatant. This was followed by three washes in PBS and centrifuging at 13,000 rpm at 4 °C for 5 min in between each wash. Cells were quenched in liquid nitrogen with 1 mL of 2:2:1 acetonitrile/methanol/water. Freeze/thaw cycles between a water bath at 37 °C and liquid nitrogen were performed 3 times to lyse the cells. The supernatant was transferred to a glass vial and dried under 99.999% pure nitrogen flow. A bicinchoninic acid assay was performed on the cell pellet to measure protein concentration; these values were used to normalize the volume of 50:50 acetonitrile/water, which was used to reconstitute the samples for LCMS analysis. The spent media were not extracted for analysis of the external metabolites.

### 4.2. LCMS Conditions

Two separate untargeted LCMS analyses were performed on the *C. sedlakii* extracts on an Elute UHPLC and Bruker Impact II quadrupole time of flight (QTOF) LCMS system. The first analysis was performed on a Scherzo SM column (Imtakt, Portland OR, USA) in reverse phase. A gradient elution was used from 98% channel A (0.1% formic acid in water) to 98% channel B (0.1% formic acid in acetonitrile) over 22 min, using a flow rate of 200 µL/min and a 15 µL injection volume. The mass spectrometer was operated in positive mode using a mass range of 50–1000 *m*/*z* at a scan rate of 3 Hz. The capillary voltage was set to 5500 V, with nebulizer gas pressure set at 1.8 bar, dry gas flow set at 4.0 L/min, and dry gas temperature set at 200 °C. Ion funnels 1 and 2 were set to 100 and 200 V, respectively. The collision RF was set to 750 V, with a transfer time of 60 µs and a pre-pulse storage time of 5 µs.

The second analysis was performed on a Luna NH2 column (Phenomenex, Torrance CA, USA) using HILIC. A gradient elution was used from 98% channel B (acetonitrile) to 98% channel A (20 mM ammonium acetate, 40 mM ammonium hydroxide) over 55 min using a flow rate of 100 µL/min and a 15 µL injection volume. The mass spectrometer was operated in negative mode using a mass range from 50 to 1000 *m*/*z* at a scan rate of 3 Hz. The capillary voltage was set to 3500 V with nebulizer pressure of 2.0 bar, a dry gas flow of 6.0 L/min, and dry gas temperature of 200 °C. Ion funnels 1 and 2 were set to 100 and 200 V, respectively. The collision RF was set to 350 V, with a transfer time of 75 µs and a pre-pulse storage time of 5.0 µs. Data were acquired using Bruker HyStar v4.1 SR2 and converted to mzxml format using Bruker CompassXport v3.0.7.

The data obtained in this study will be accessible at the NIH Common Fund’s National Metabolomics Data Repository (supported by NIH grant, U01-097430) website, the Metabolomics Workbench, https://www.metabolomicsworkbench.org.

### 4.3. XCMS Online Feature Detection

XCMS Online was used to perform feature detection, retention time correction, and to identify statistically significant features that changed between mid to late exponential phases and between late exponential and stationary phases. XCMS Online is a freely available metabolomics data processing and analysis software (xcmsonline.scripps.edu) [14,24,28,29]. Parameters used for feature detection, retention time alignment, and statistical analysis are described in Appendix A.

### 4.4. PyFBA Flux Balance Analysis

PyFBA [21] is the flux balance analysis python script used to generate the metabolic model from the genome sequence and the predicted metabolite list. This software is freely available at http://linsalrob.github.io/PyFBA/. Details on how to generate a PyFBA metabolite list from an annotated genome can be found as previously described. Any flux balance analysis software can be used, provided that the metabolite list is in the proper format described at https://github.com/mmarney/MS_FBA.

### 4.5. Metabolite Prediction for Metabolomics Integration

The metabolite prediction against the flux balance model was performed using a script developed in-house named MS_FBA. MS_FBA is an R script available for download from https://github.com/mmarney/MS_FBA and can be run on any standard computer with R installed. For details on installation and implementation of MS_FBA, see https://github.com/mmarney/MS_FBA/blob/master/INSTALLATION.md. Details on script parameters are available in the Appendix A.

## Figures and Tables

**Figure 1 metabolites-10-00156-f001:**
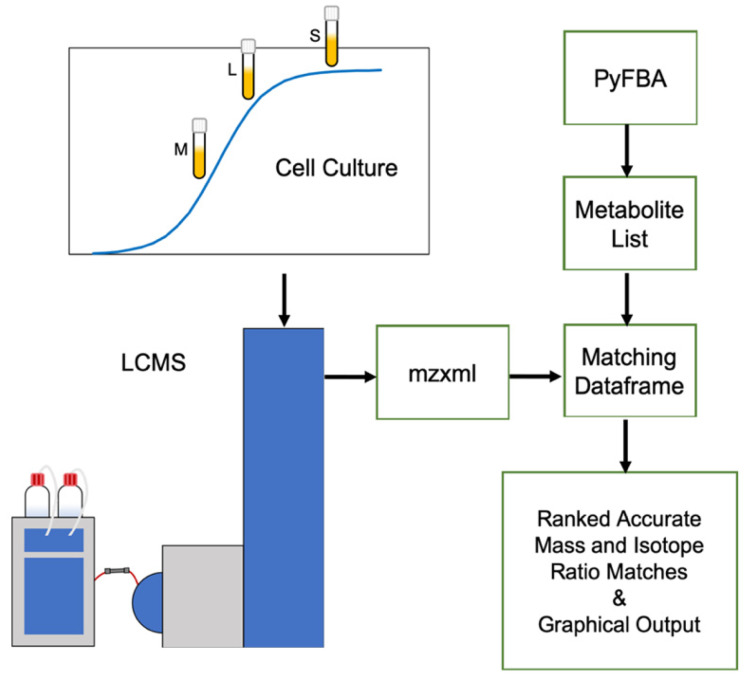
MS_FBA workflow illustrating the necessary input files and algorithmic processes involved in integrating metabolomics data with genome-based metabolic models. The algorithm requires two sets of data: one from mid (M), late (L) and stationary (S) phase cell culture extracts that have been analyzed using untargeted liquid chromatography mass spectrometry (LCMS) that have been converted to mzxml file format; the other is a metabolite list generated from flux balance analysis software, in this case we use PyFBA. The *m/z* features are searched within the metabolite list based on accurate mass and isotope ratios, then output as a ranked list and graphical visualization.

**Figure 2 metabolites-10-00156-f002:**
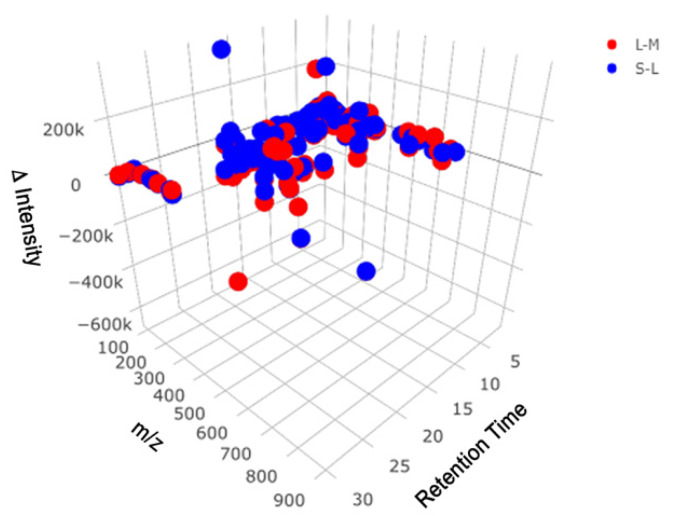
A 3D graphical output from MS_FBA showing features with potential metabolite matches between significant features from reverse phase data and the PyFBA predicted metabolites list. There are two data points for every feature: the intensity difference between late-log phase and mid-log phase sample classes (**red**), and the intensity difference between the stationary phase and late-log phase sample classes (**blue**).

**Figure 3 metabolites-10-00156-f003:**
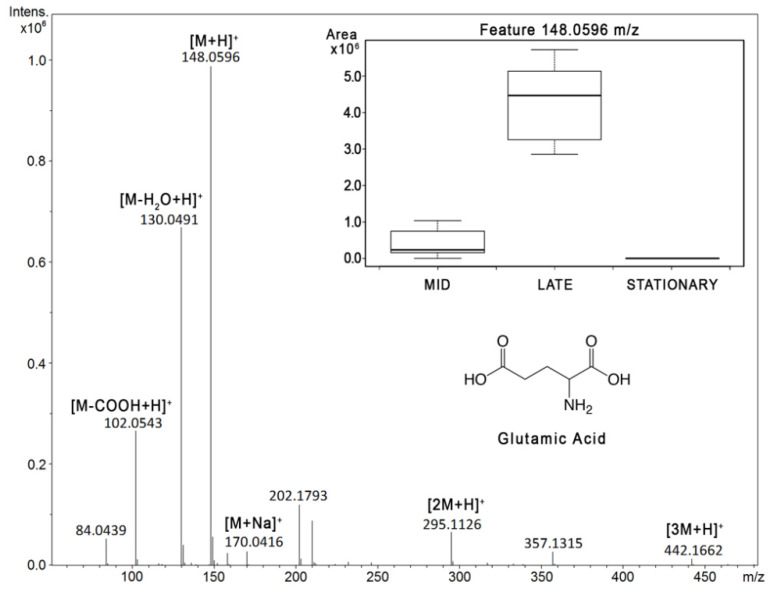
Mass spectrum taken from late stationary phase sample L5, depicting the metabolite feature eluting at 1.9 min correlating to glutamic acid and [M+H]^+^ at 148.0596 *m*/*z*. Identified in the spectrum are the loss of carboxylic acid [M-COOH+H]^+^, the loss of water [M-H_2_O+H]^+^, the sodium adduct [M+Na]^+^, the dimer [2M+H]^+^, and the trimer [3M+H]^+^. The inset graph shows the production and decline of the feature identified as glutamic acid for mid, late, and stationary phases, with box and whisker plots from the five biological replicates.

**Figure 4 metabolites-10-00156-f004:**
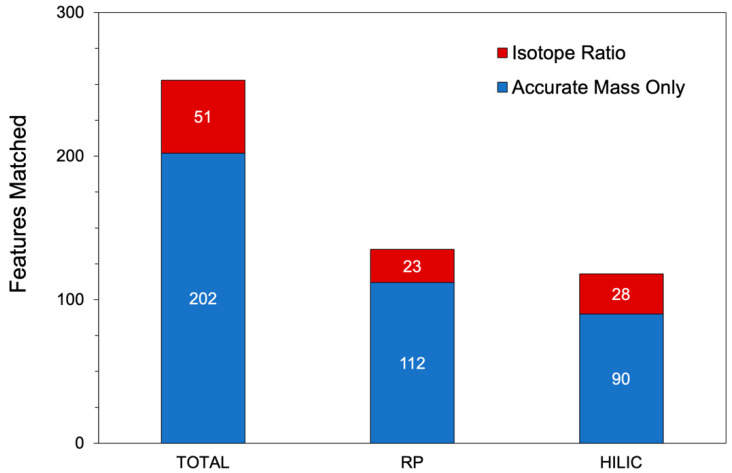
A summary of metabolite features MS_FBA-matched to the predicted metabolites generated from the PyFBA metabolic model with either accurate mass only (**blue**) or isotope ratio (**red**). The total column is the combination of both RP positive mode and HILIC negative runs, without excluding duplicate features between polarities.

**Table 1 metabolites-10-00156-t001:** Summary of metabolite features matched to reverse phase (RP) and hydrophilic interaction liquid chromatogprahy (HILIC) data using MS_FBA.

Features Matched	RP	HILIC
XCMS features (pre-MS_FBA)	1210	1560
Matched to PyFBA	135	118
PyFBA isotope matches	23	28
Matched to Model SEED	846	621
Model SEED isotope matches	107	90
Unique annotated metabolites to PyFBA	218	
PyFBA compounds in search list	699	
Model SEED compounds in search list	27,693	

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
