# Peer review of "Towards Predicting Gut Microbial Metabolism: Integration of Flux Balance Analysis and Untargeted Metabolomics"

_metabolites, 2020, doi:10.3390/metabo10040156_

Round 1

Reviewer 1 Report

This manuscript describes the measurement by MS of a list of metabolite masses by Citrobacter sedlakii growing in a minimal medium with 0.2% glucose and introduces a programme, MS_FBA, to correlate the MS data with the predicted genome model using PyFBA. The authors suggest that MS_FBA could be used to indicate the validity of the gap filling approach by combining experimental MS data with online genome databases.

This is an interesting approach which could have utility in validating genome-based gap-filling predictions and make a contribution to the expanding field of metabolomics.

It may be that more work is required to improve the matching of measured metabolite masses with output. I am unclear why 0.2% glucose was chosen for the baseline of these experiments. Output by C. sedlakii in response to a greater range of glucose concentrations is required to give a complete picture and fuller validation. Before these experiments are performed, it would be difficult to justify the authors’ ambitions to apply the findings to the gut microbiome.

Growing the cells in minimal medium clearly produces a low cell density which further complicates the evaluation of metabolite extraction and further experiments with higher cell densities would also contribute to producing a clearer picture.

As the technique stands at present, I am unclear on the contribution made and feel that more data is required before the approach will have sufficient utility.

Author Response

April 7, 2020

Dear Editor,

Thank you for sending us the Reviewer’s detailed critique of the manuscript entitled “Predicting Gut Microbial Metabolism: Integration of Flux Balance Analysis and Untargeted Metabolomics.” We have taken their suggestions, implemented and/or responded to each point. Please find our responses outlined in blue below.

Yours sincerely,

Erica Forsberg

Reviewer 1

This manuscript describes the measurement by MS of a list of metabolite masses by Citrobacter sedlakii growing in a minimal medium with 0.2% glucose and introduces a programme, MS_FBA, to correlate the MS data with the predicted genome model using PyFBA. The authors suggest that MS_FBA could be used to indicate the validity of the gap filling approach by combining experimental MS data with online genome databases.

This is an interesting approach which could have utility in validating genome-based gap-filling predictions and make a contribution to the expanding field of metabolomics.

It may be that more work is required to improve the matching of measured metabolite masses with output. I am unclear why 0.2% glucose was chosen for the baseline of these experiments. 

Thank you for pointing out this issue. The media choice was based on the original plate-based assay that used 0.2% as in “Cuevas, D.A.; et al. Elucidating genomic gaps using phenotypic profiles. F1000Research 2016, 3, 210.” The scale up was lower than anticipated and in future work we will be using a minimum of 2% glucose or full media models such as LB. However, in this instance, the proof of concept is still demonstrated on the 0.2% data with lower matching. This has been updated in the methods and results and discussion section to reflect that.

Output by C. sedlakii in response to a greater range of glucose concentrations is required to give a complete picture and fuller validation. Before these experiments are performed, it would be difficult to justify the authors’ ambitions to apply the findings to the gut microbiome.

This is an important criticism the reviewer addresses. Here we are giving a simple example with only one nutrient source. As we are working with a large dataset and simultaneously comparing with a predicted model, we needed to begin using smaller model to validate our process before going to a more complex system. We have addressed this in the introduction of the manuscript.

Growing the cells in minimal medium clearly produces a low cell density which further complicates the evaluation of metabolite extraction and further experiments with higher cell densities would also contribute to producing a clearer picture.

As the technique stands at present, I am unclear on the contribution made and feel that more data is required before the approach will have sufficient utility.

We agree with the reviewer that more complex systems with higher cell densities are needed.  However, the major scientific contribution is the development of the R script to perform the comparison.  Even though the metabolomics data is of low signal intensity, the script still functions effectively and the results lead to the next steps we must take to improve.  This also shows the utility of the script in extremely low abundance situations, to which it actually performs quite well considering the conditions. This has been clarified in the discussion portion of the manuscript.

Reviewer 2 Report

In this article, the authors have stated that they attempted to generate a list of accurate metabolite masses produced from the human commensal bacteria Citrobacter sedlakii growth in the presence of the simple carbon source glucose using untargeted mass spectrometry-based metabolomics data. This is a very ambiguous and confusing manuscript that lacks scientific soundness and novel findings. The manuscript, which was focusing in bioinformatics analysis, is hard to read and follow because of containing many grammar errors and no obvious hypotheses that would be addressed. Therefore, it is likely beyond the scope to publish in the Metabolites.

Here are specific comments:

  1. There was no statement for the specific hypothesis and aims for this study in the manuscript.
  2. In Materials and Methods section, the Bacterial Culturing did not provide any information about the condition for culturing Citrobacter sedlakii at all. The authors just mentioned “we grew C. sedlakii as a model system on which the metabolomics workflow was performed” in line 193. What does this statement mean? Also, why only minimal media was utilized in this study? Diverse media should be conducted to compare the discrepancy of microbial metabolism under different environmental exposure.
  3. The authors claimed that “biological replicates of C. sedlakii cultures were prepared” in line 194. However, there was no statement for the information regarding the raw data of LCMS that were deposited to the public repository and were available to access with a unique accession number.
  4. Which type of targeted metabolites, extracellular secreted from the bacterial cells or internal within the bacterial cells, that authors would like to investigate in this study?
  5. The important question is which novel insights that we would learn from this prediction. Because this model is still far from or even opposite to the gastrointestinal tract that would give a closer understanding of the gut microbial metabolism.

Author Response

April 7, 2020

Dear Editor,

Thank you for sending us the Reviewer’s detailed critique of the manuscript entitled “Predicting Gut Microbial Metabolism: Integration of Flux Balance Analysis and Untargeted Metabolomics.” We have taken their suggestions, implemented and/or responded to each point. Please find our responses outlined in blue below.

Yours sincerely,

Erica Forsberg

Reviewer 2

In this article, the authors have stated that they attempted to generate a list of accurate metabolite masses produced from the human commensal bacteria Citrobacter sedlakii growth in the presence of the simple carbon source glucose using untargeted mass spectrometry-based metabolomics data. This is a very ambiguous and confusing manuscript that lacks scientific soundness and novel findings. The manuscript, which was focusing in bioinformatics analysis, is hard to read and follow because of containing many grammar errors and no obvious hypotheses that would be addressed. Therefore, it is likely beyond the scope to publish in the Metabolites.

Here are specific comments:

  1. There was no statement for the specific hypothesis and aims for this study in the manuscript.

Thank you for the suggestion, we have included a more specific hypothesis (lines 50-52, 80-82) and detailed aims for this study in the introduction (lines 60-66).

  1. In Materials and Methods section, the Bacterial Culturing did not provide any information about the condition for culturing Citrobacter sedlakii at all. The authors just mentioned “we grew C. sedlakii as a model system on which the metabolomics workflow was performed” in line 193. What does this statement mean? Also, why only minimal media was utilized in this study? Diverse media should be conducted to compare the discrepancy of microbial metabolism under different environmental exposure.

Thank you for the suggestion. We have expanded and clarified the methods for bacterial culturing and added the reasoning for the use of 0.2% glucose in the manuscript. We agree with the reviewer that the next logical step is to test the script with a more complex media to validate the metabolic model. This will be a followup study and has been discussed in the Discussion section.

  1. The authors claimed that “biological replicates of C. sedlakii cultures were prepared” in line 194. However, there was no statement for the information regarding the raw data of LCMS that were deposited to the public repository and were available to access with a unique accession number.

We have made our data available through Metabolomics Workbench. The publicly accessible project ID and link will be generated following review of the project. The project’s current data tracking ID is DataTrackID1982.

  1. Which type of targeted metabolites, extracellular secreted from the bacterial cells or internal within the bacterial cells, that authors would like to investigate in this study?

The targeted metabolites are those predicted in the metabolic model. Instead of performing targeted analysis on a list of hundreds of potential metabolites, MS_FBA uses untargeted metabolomics data. We have now included line 60-64 in the introduction to address this.  

Currently, the workflow only looks at internal metabolites, as this is specific to our extraction protocol. Any set of untargeted metabolomics data can be used to compare with a metabolic model. This is mentioned now in line 65-66 in the introduction. Our goal is to analyze extracellular components in the media alongside the intracellular components. However, for this preliminary study where we develop the script, we used a smaller dataset.

  1. The important question is which novel insights that we would learn from this prediction. Because this model is still far from or even opposite to the gastrointestinal tract that would give a closer understanding of the gut microbial metabolism.

We are not attempting to model the gastrointestinal tract with this study, but we are developing a means to analyze and predict gut bacterial metabolism. In order to improve our understanding of metabolic pathway activation, we need to be able to predict whether metabolites are produced or consumed. After the development of this script, we will move to the next stage of applying it to more complex mixtures, but we need to start with a simpler model. This has been clarified in the discussion.

The ModelSEED database comparison generated many more matches than the predicted model indicating the gap filling methods used were not sufficient. We have learned that we can improve our gap filling methods by using untargeted metabolomics data, and this will be the topic of a future manuscript. In addition, we are hoping to use these methods for predicting metabolite production in bacteria using more complex media conditions and multiple bacterial species. However, we needed to develop the script with a simple model first - hence the glucose minimal media. This has been clarified in the text.

Reviewer 3 Report

Summary:

In this article the authors have developed an R script that matches untargeted metabolomics data to metabolites in a metabolic model (or another database such as ModelSEED). Use of the script is presented for a commensal gut microbe, Citrobacter sedlakii. Untargeted metabolomics data were collected at three different growth phases and analyzed with two different methods, reverse phase and hydrophilic interaction liquid chromatography. Through use of the developed matching method, 218 of 699 (or 31%) of metabolites contained in the metabolic model were matched with the untargeted metabolomics data, while more than 1400 additional metabolites from the collected data set were matched to compounds in the ModelSEED database. While the topic of metabolomics-informed flux balance analysis holds promise and is of interest to the scientific community, I have several substantial concerns regarding the method details, presentation of data, and appropriate framing of the study in context with the literature that prevent me from recommending publication in the present form.

Major Concerns:

  1. The introduction lacks a clear and compelling goal/purpose explicitly stating the impact of this article to the scientific community; much of the introduction (especially paragraph 2) simply states the work that was done rather than placing it into context. Specifically, a rationale explaining the advantage of using untargeted rather than targeted metabolomics data should be included. A recent conference panel report (More, Tushar, et al. "Metabolomics and its integration with systems biology: PSI 2014 conference panel discussion report." Journal of Proteomics 127 (2015): 73-79.) has posed that non-targeted metabolomics presents significant challenges for integration with systems biology, namely absolute quantification and identification of compounds, as well as bias toward high-abundance molecules. These limitations and challenges are not thoroughly addressed in the article. As it stands, it is unclear to me how the presented approach advances modeling over other previous solutions/tools, such as the JBEI quantitative metabolic modeling library (Birkel, Garrett W., et al. "The JBEI quantitative metabolic modeling library (jQMM): A python library for modeling microbial metabolism." BMC Bioinformatics 18.1 (2017): 205.) which would integrate more easily with PyFBA.
  2. One of the major issues I find in the article is the presentation of data; the figures are not informative, well explained/interpreted, or professionally formatted. The three-dimensional plots in Figures 2 and 3 are very basic; additional formatting should be done to prepare them for publication (importantly, units in Figure 2 and axis titles and units in Figure 3). Additionally, data series colors are inconsistent and unclear: Figure 2 caption states that the data series denoting stationary phase & late-log phase difference is red, but the legend shows it as a blue dot. The captions for Figures 2 and 3 also switch red and blue for different data series, leading to confusion. In Figure 3, there appears to be only one clear visual difference between plots A and B (an extra blue point towards the upper left of plot A that is not in plot B); it seems unnecessary to show both if they are so similar. Most critically, while it is nice to have the visuals of the three-dimensional plots, the results and discussion do not extract any meaning from these graphs other than to state “this is the graphical output”. If these graphs are to serve as the central figures for the article, this needs to be significantly expanded, for example by including discussion of the differences between the growth phases sampled.
  3. Line 69 cites well-established models for C. sedlakii (references # 4 and 12), yet it appears in line 75 that a new model was constructed in the current work with PyFBA. If the model is from the reference, this point should be made explicit and additional salient details about the model structure should be included. If not, an explanation should be included for why a new model was generated rather than using an existing model and more details of model construction need to be included (e.g., genome accession used, substrate components considered, biomass composition reaction, how (or if) gap-filling was performed, etc.).
  4. The authors acknowledge a potentially high false discovery rate with five biological replicates (lines 161-162); it may be insightful and worthwhile to use a more stringent p-value such as 0.01 to lower the possibility of false positives.
  5. The authors mention that low cell density in the samples was not optimal for metabolite extractions (line 173); why not use a greater culture volume or combine multiple samples to obtain more material to eliminate this issue? Methods do not include the volume of culture that was sampled. The authors recommend optimizing this portion of the experimental procedure (lines 175-176) but do not provide any suggestions on how to proceed; specific improvements should be included.
  6. Materials and Methods section 4.1 does not give any details on how the cultures were physically grown in the laboratory; line 194 only states that they were prepared with reference to an article that is a modeling paper (reference #10) and does not contain experimental details. The reference should be changed to appropriately represent the statement, and for reproducibility purposes, at the very minimum a brief statement regarding where the strain of C. sedlakii was obtained, composition and preparation of the medium, and description of the culturing conditions (e.g., temperature, pH, agitation, etc.) should be included.
  7. I consider the article to be over-promising based on its title; from the methods and results it does not appear that any actual flux distribution predictions were made; it is rather an accuracy assessment comparing metabolites included in a metabolic model with an experimental untargeted metabolomics data set. In the presented results, only 218 of 699 metabolites included in the metabolic model (or about 31%) were matched by the metabolomics data; the authors should provide some context of what this means in terms of metabolic modeling in general and flux balance analysis in particular. With a large number of metabolites showing matches to the ModelSEED database, the authors recommend in lines 185-188 that the next steps should be reiterating the metabolic model to include reactions that produce metabolites observed and additional MS/MS validation. To fulfill the article’s title of predicting gut microbial metabolism, it seems reasonable that this paper should at least go through the model reiteration step and demonstrate the differences that metabolomics refinement has on predicted flux distributions. I believe this would greatly enhance the impact of the article.

Minor Concerns:

  1. The introduction highlights intercellular communication and immune response mechanisms as an area of importance for this work (line 34), but no mention of application or extension of the presented work to interspecies interactions is included in the discussion. If the aim of the work is truly to predict gut microbial metabolism, then microbial community interactions and environmental context should be discussed.
  2. The difference between reverse phase and HILIC LC-MS methods and the rationale for using both of these methods is unclear initially; all that is mentioned is enhancement of metabolite coverage (lines 166-167). This point should be expanded and clarified for non-metabolomics experts.
  3. I am concerned about how the concept of gap-filling a metabolic model is portrayed in lines 181 and 182, to “provide all the additional reactions first, then remove them until the predicted biomass matches the phenotypic growth curve.” No references for gap-filling were cited in the article. In the conventional sense, gap-filling is used to complete connectivity of networks, for example in the case of mis-annotation where a reaction in a pathway is found lacking. Gaps are filled based on literature research, phylogenetic comparisons, or biochemical knowledge, to fulfill functionality in the model (e.g., biomass synthesis or another particular physiological function). Thiele and Palsson’s Nature Protocols article contains a thorough description (Thiele, Ines, and Bernhard Ø. Palsson. "A protocol for generating a high-quality genome-scale metabolic reconstruction." Nature Protocols 5.1 (2010): 93.).

Author Response

April 7, 2020

Dear Editor,

Thank you for sending us the Reviewer’s detailed critique of the manuscript entitled “Predicting Gut Microbial Metabolism: Integration of Flux Balance Analysis and Untargeted Metabolomics.” We have taken their suggestions, implemented and/or responded to each point. Please find our responses outlined in blue below.

Yours sincerely,

Erica Forsberg

Reviewer 3

In this article the authors have developed an R script that matches untargeted metabolomics data to metabolites in a metabolic model (or another database such as ModelSEED). Use of the script is presented for a commensal gut microbe, Citrobacter sedlakii. Untargeted metabolomics data were collected at three different growth phases and analyzed with two different methods, reverse phase and hydrophilic interaction liquid chromatography. Through use of the developed matching method, 218 of 699 (or 31%) of metabolites contained in the metabolic model were matched with the untargeted metabolomics data, while more than 1400 additional metabolites from the collected data set were matched to compounds in the ModelSEED database. While the topic of metabolomics-informed flux balance analysis holds promise and is of interest to the scientific community, I have several substantial concerns regarding the method details, presentation of data, and appropriate framing of the study in context with the literature that prevent me from recommending publication in the present form.

Major Concerns:

  1. The introduction lacks a clear and compelling goal/purpose explicitly stating the impact of this article to the scientific community; much of the introduction (especially paragraph 2) simply states the work that was done rather than placing it into context. Specifically, a rationale explaining the advantage of using untargeted rather than targeted metabolomics data should be included. A recent conference panel report (More, Tushar, et al. "Metabolomics and its integration with systems biology: PSI 2014 conference panel discussion report." Journal of Proteomics 127 (2015): 73-79.) has posed that non-targeted metabolomics presents significant challenges for integration with systems biology, namely absolute quantification and identification of compounds, as well as bias toward high-abundance molecules. These limitations and challenges are not thoroughly addressed in the article.

We understand the reviewers concerns about untargeted metabolomics data. Untargeted data is easier to produce data and then analyze quickly using bioinformatics. This is not a quantitative approach, this is a yes/no scenario to see how the metabolome is changing on a global scale compared to the predictive model. We have mentioned this important use of untargeted data in the introduction. Data analysis, particularly of low abundance molecules has improved significantly over the past 5 years and we have also included references and more detailed information on this in the introduction.

As it stands, it is unclear to me how the presented approach advances modeling over other previous solutions/tools, such as the JBEI quantitative metabolic modeling library (Birkel, Garrett W., et al. "The JBEI quantitative metabolic modeling library (jQMM): A python library for modeling microbial metabolism." BMC Bioinformatics 18.1 (2017): 205.) which would integrate more easily with PyFBA.

Thank you for this point. Novelty is truly in the comparison of untargeted metabolomics data with the predicted metabolic model. This is advantageous because of the simplicity of the system. We can quickly look at metabolite production and compare it with predicted the model without the need for isotopically labeled precursors. Targeted analysis would not address compounds that are not predicted in the model, thereby fixing gap filling and gene annotation issues. We have clarified this in the introduction section.

  1. One of the major issues I find in the article is the presentation of data; the figures are not informative, well explained/interpreted, or professionally formatted. The three-dimensional plots in Figures 2 and 3 are very basic; additional formatting should be done to prepare them for publication (importantly, units in Figure 2 and axis titles and units in Figure 3). Additionally, data series colors are inconsistent and unclear: Figure 2 caption states that the data series denoting stationary phase & late-log phase difference is red, but the legend shows it as a blue dot. The captions for Figures 2 and 3 also switch red and blue for different data series, leading to confusion. In Figure 3, there appears to be only one clear visual difference between plots A and B (an extra blue point towards the upper left of plot A that is not in plot B); it seems unnecessary to show both if they are so similar. Most critically, while it is nice to have the visuals of the three-dimensional plots, the results and discussion do not extract any meaning from these graphs other than to state “this is the graphical output”. If these graphs are to serve as the central figures for the article, this needs to be significantly expanded, for example by including discussion of the differences between the growth phases sampled.

Thank you for this constructive criticism of our manuscript. We have replaced Figure 3 to better interpret the results, and moved the original Figure 3 to Supplementary Materials. Major changes were made to both the results and discussion section to address these issues. We hope the improved formatting on the new Figure 3 is more in line with standards. We have also corrected the Figure 2 caption for the metabolite features matched to the predicted metabolite list; thank you for catching that, it was incorrect in the original version. 

  1. Line 69 cites well-established models for sedlakii (references # 4 and 12), yet it appears in line 75 that a new model was constructed in the current work with PyFBA. If the model is from the reference, this point should be made explicit and additional salient details about the model structure should be included. If not, an explanation should be included for why a new model was generated rather than using an existing model and more details of model construction need to be included (e.g., genome accession used, substrate components considered, biomass composition reaction, how (or if) gap-filling was performed, etc.).

Thank you for bringing this to our attention. The C. sedlakii gene annotation and model are not new, but the metabolite list was generated by our lab from that model using PyFBA. Here we use the metabolite list as our metabolic model as it contains the predicted compounds we are considering in our metabolomics data. More details on the set of linear equations usd to derive biomass composition reactions, and more details on the gap filling process are provided in references “Cuevas, D.A.; Edwards, R.A. PMAnalyzer: a new web interface for bacterial growth curve analysis. Bioinformatics 2017, 33, 1905–1906” and “Cuevas, D.A.; Edirisinghe, J.; Henry, C.S.; Overbeek, R.; O’Connell, T.G.; Edwards, R.A. From DNA to FBA: How to Build Your Own Genome-Scale Metabolic Model. Front. Microbiol. 2016, 7, 907”. 

  1. The authors acknowledge a potentially high false discovery rate with five biological replicates (lines 161-162); it may be insightful and worthwhile to use a more stringent p-value such as 0.01 to lower the possibility of false positives.

Thank you for pointing this out and yes we agree that in general a P-value <0.01 is more standard for metabolomics data. A more stringent p-value of 0.01 will be used in future analyses with the data generated from growth in 2% glucose and LB media. The less stringent p-value of 0.05 was to compensate for the low signal intensity of the 0.2% glucose data.

  1. The authors mention that low cell density in the samples was not optimal for metabolite extractions (line 173); why not use a greater culture volume or combine multiple samples to obtain more material to eliminate this issue? Methods do not include the volume of culture that was sampled. The authors recommend optimizing this portion of the experimental procedure (lines 175-176) but do not provide any suggestions on how to proceed; specific improvements should be included.

Thank you for pointing out this shortcoming. A detailed description of culturing conditions has been added, along with justification for the chosen conditions and specific improvements on how to do so have been included.

  1. Materials and Methods section 4.1 does not give any details on how the cultures were physically grown in the laboratory; line 194 only states that they were prepared with reference to an article that is a modeling paper (reference #10) and does not contain experimental details. The reference should be changed to appropriately represent the statement, and for reproducibility purposes, at the very minimum a brief statement regarding where the strain of sedlakii was obtained, composition and preparation of the medium, and description of the culturing conditions (e.g., temperature, pH, agitation, etc.) should be included.

As mentioned in the previous critique, a detailed description of culturing conditions has been added, along with justification for the chosen conditions.

  1. I consider the article to be over-promising based on its title; from the methods and results it does not appear that any actual flux distribution predictions were made; it is rather an accuracy assessment comparing metabolites included in a metabolic model with an experimental untargeted metabolomics data set. In the presented results, only 218 of 699 metabolites included in the metabolic model (or about 31%) were matched by the metabolomics data; the authors should provide some context of what this means in terms of metabolic modeling in general and flux balance analysis in particular. With a large number of metabolites showing matches to the ModelSEED database, the authors recommend in lines 185-188 that the next steps should be reiterating the metabolic model to include reactions that produce metabolites observed and additional MS/MS validation. To fulfill the article’s title of predicting gut microbial metabolism, it seems reasonable that this paper should at least go through the model reiteration step and demonstrate the differences that metabolomics refinement has on predicted flux distributions. I believe this would greatly enhance the impact of the article.

We agree completely with the reviewer that the reiteration step is the next critical phase in this project. However, this project took a significant amount of work to reach the first completed phase of performing the preliminary validation step of the metabolic model using untargeted metabolomics data. Although FBA does an excellent job at predicting growth under different nutrient conditions, there is no way to validate these results without a significant amount of quantitative targeted analysis. This is why we chose to take advantage of the breadth of information available in untargeted data by comparison with the ModelSEED database. To build the next algorithm would take many months of sustained effort that we believe would not show the impact of the results already achieved.

Minor Concerns:

  1. The introduction highlights intercellular communication and immune response mechanisms as an area of importance for this work (line 34), but no mention of application or extension of the presented work to interspecies interactions is included in the discussion. If the aim of the work is truly to predict gut microbial metabolism, then microbial community interactions and environmental context should be discussed.

Thank you for this constructive criticism. We have addressed this in the discussion as to our aims of the script and how we plan to move forward in applying it to future studies.

  1. The difference between reverse phase and HILIC LC-MS methods and the rationale for using both of these methods is unclear initially; all that is mentioned is enhancement of metabolite coverage (lines 166-167). This point should be expanded and clarified for non-metabolomics experts.

Thank you for pointing this out. We have added a description as to why this is important and included a reference for further explanation “Ortmayr, K.; Causon, T.J.; Hann, S.; Koellensperger, G. Increasing selectivity and coverage in LC-MS based metabolome analysis. TrAC - Trends Anal. Chem. 2016, 82, 358–366”.

  1. I am concerned about how the concept of gap-filling a metabolic model is portrayed in lines 181 and 182, to “provide all the additional reactions first, then remove them until the predicted biomass matches the phenotypic growth curve.” No references for gap-filling were cited in the article. In the conventional sense, gap-filling is used to complete connectivity of networks, for example in the case of mis-annotation where a reaction in a pathway is found lacking. Gaps are filled based on literature research, phylogenetic comparisons, or biochemical knowledge, to fulfill functionality in the model (e.g., biomass synthesis or another particular physiological function). Thiele and Palsson’s Nature Protocols article contains a thorough description (Thiele, Ines, and Bernhard Ø. Palsson. "A protocol for generating a high-quality genome-scale metabolic reconstruction." Nature Protocols 1 (2010): 93.).

Thank you for the critique on our discussion of gap filling. Details on gap filling are discussed in great detail in the PyFBA paper.  However, a more thorough discussion of this and the importance compared to the untargeted metabolomics data has been included in the discussion.

Round 2

Reviewer 1 Report

The revised manuscript makes a good case for the use of this approach which combines MS data with predicted genome modelling to analyse output of metabolites by C sedlakii. The authors have done a good job in expanding on their favoured methods and indicating what role could be played by this approach in predicting behaviour of the gut microbiome.

I suspect the title could be more easily justified if it were amended to :

Towards Predicting Gut Microbial Metabolism: Integration of 2Flux Balance Analysis and Untargeted Metabolomics

Otherwise, I am persuaded that this ms constitutes a good contribution to the field.

L46 : “exist” should be “exists”.

Author Response

Dear Reviewer,

Thank you for your time and efforts in reviewing the manuscript originally entitled "Predicting Gut Microbial Metabolism: Integration of Flux Balance Analysis and Untargeted Metabolomics." We have modified the document based on your most recent review. Your constructive critique of this research has ensured the quality of scientific work being published and I really appreciate it. Please see responses to your points below in blue.

Yours sincerely,

Erica Forsberg

I suspect the title could be more easily justified if it were amended to :

Towards Predicting Gut Microbial Metabolism: Integration of 2Flux Balance Analysis and Untargeted Metabolomics

Otherwise, I am persuaded that this ms constitutes a good contribution to the field.

Thank you for these comments. We agree with the amendment to the title and it has been updated.

L46 : “exist” should be “exists”.

Thank you for pointing this out. It has been corrected.

Reviewer 2 Report

Many thank for the efforts that the authors have invested in the revised version. The authors have nicely addressed all the comments raised by my previous review report. The quality of the revised manuscript was greatly improved. Thus, it is ready to publish on Metabolites.

Author Response

Dear Reviewer,

Thank you for your time and efforts in reviewing our manuscript entitled "Predicting Gut Microbial Metabolism: Integration of Flux Balance Analysis and Untargeted Metabolomics." We appreciate your feedback as it has significantly improved the scientific integrity of the presentation of this research.

Your sincerely,

Erica Forsberg

Reviewer 3 Report

Thank you to the authors for taking the suggestions into account and thoroughly editing the manuscript. The revisions clarify my original questions and concerns about the study, and I believe will more effectively communicate the work to a broader audience and ensure reproducibility. I see just a few minor points for clarification:

  • In the first paragraph of the discussion, some of the newly added text is difficult to follow: “In the case of glutamic acid where not all neutral losses were identified, our blanket adduct list cannot be utilized to identify all neutral losses since not all metabolites will not have the same functional groups…” (lines 186-188). The multiple instances of not/cannot make it hard to discern the meaning.
  • Forsberg et al., Nature Protocols, 2018 citation is listed twice in the references (#14 and 29).

Author Response

Dear Reviewer,

We appreciate your comments on the revised version of "Predicting Gut Microbial Metabolism: Integration of Flux Balance Analysis and Untargeted Metabolomics."  Your feedback and suggestions have been very helpful in ensuring the quality of this manuscript. We have updated the manuscript to address your concerns. Thank you for taking the time to review this work. I have responded to your points below in blue.

Yours Sincerely,

Erica Forsberg

Reviewer 3:

In the first paragraph of the discussion, some of the newly added text is difficult to follow: “In the case of glutamic acid where not all neutral losses were identified, our blanket adduct list cannot be utilized to identify all neutral losses since not all metabolites will not have the same functional groups…” (lines 186-188). The multiple instances of not/cannot make it hard to discern the meaning.

Thank you for your comments on this newly written section. We have clarified the text to read "In the case of glutamic acid (Figure 3), not all neutral losses were identified in the peak group.  Along with the [M+H]+ peak, MS_FBA correctly identified the water neutral loss, the sodium adduct and the dimer.  It did not detect the loss of the carboxylate group or the trimer.  The adduct list (Supplemental Table S3) could easily be adjusted to include the trimer, however, to identify all neutral losses with a single adduct list is not feasible since metabolites have different functional groups susceptible to in-source fragmentation."

Forsberg et al., Nature Protocols, 2018 citation is listed twice in the references (#14 and 29).

Thank you for pointing out this duplicate reference; it has been removed.